# Training Task Experts through Retrieval Based Distillation

## Abstract

One of the most reliable ways to create deployable models for specialized tasks is to obtain an adequate amount of high-quality task-specific data. However, for specialized tasks, often such datasets do not exist. Existing methods address this by creating such data from large language models (LLMs) and then distilling such knowledge into smaller models. However, these methods are limited by the quality of the LLMs output, and tend to generate repetitive or incorrect data. In this work, we present **Re**trieval **Base**d Distillation (ReBase), a method that first retrieves data from rich online sources and then transforms them into domain-specific data. This method greatly enhances data diversity. Moreover, ReBase generates Chain-of-Thought reasoning and distills the reasoning capacity of LLMs. We test our method on 4 benchmarks and results show that our method significantly improves performance by up to **7.8%** on SQuAD, **1.37%** on MNLI, and **1.94%** on BigBench-Hard. [1]

## 1 Introduction

How can we effectively obtain high-quality models for specific tasks? Large Language Models (LLMs) have shown impressive generalization abilities and can, to some extent, perform specific tasks using only the task instructions and few-shot in-context examples (OpenAI, 2023; Bubeck et al., 2023; AI@Meta, 2024). However, these models can contain tens or hundreds of billions of parameters, making them computationally expensive to use in practice, and in many cases these models underperform smaller models fine-tuned on task-specific data (Mosbach et al., 2023; Viswanathan et al., 2023b; Bertsch et al., 2024).

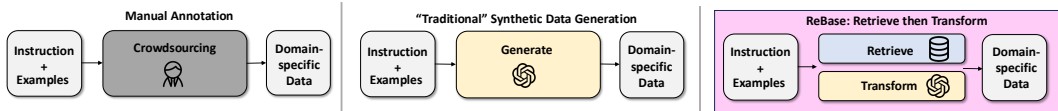

Figure 1: **Motivation of ReBase.** Previous methods either uses manually annotated data or use LLMs to generate synthetic data. This is either too costly or lacks diversity/quality. ReBase retrieves data from existing examples then uses an LLM to create new domain-specific data based on the retrieved content.

One bottleneck to creating such fine-tuned models is the lack of large corpora of task-specific data (Villalobos et al., 2022; Zhao et al., 2024). Therefore, a key issue for this problem is how to obtain adequate high quality data that meets the user's need. Recent works have used distillation from LLMs to generate synthetic training data (Ye et al., 2022b;a; Gao et al., 2023; Jung et al., 2024; Viswanathan et al., 2023a; Yu et al., 2023a; He et al., 2023; Hu et al., 2024; Honovich et al., 2022; Xiao et al., 2024; Chen et al., 2024; Yu et al., 2023b; Wang et al., 2023a). These methods use the user's instruction and a small number of in-context examples as the prompt to let LLMs generate labeled, domain-specific data. These data are then used to finetune the models to be deployed. Such methods have shown potential to improve a small model's ability to follow a specific set of instructions. However, these methods often suffer from diversity issues: the generated examples tend to

---

[1]Code Available at https://github.com/Anonymous00127/ReBase

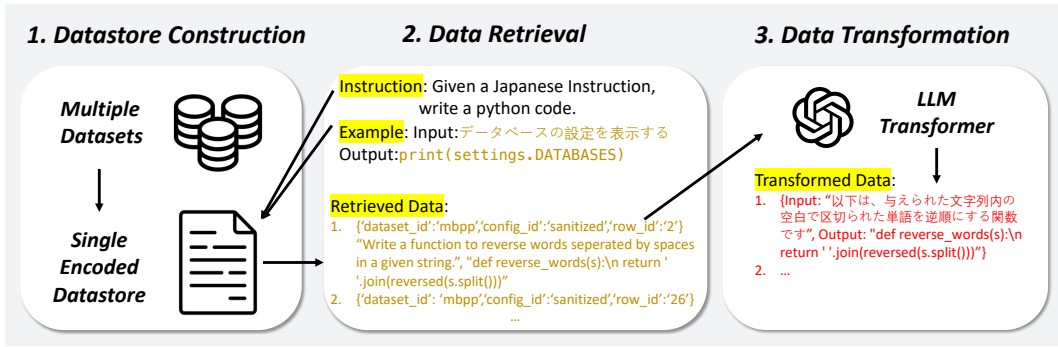

Figure 2: **Pipeline of ReBase.** First, ReBase iterates over a large number of datasets available on Hugging Face Datasets and encodes each item in this datasets to build a large datastore. Then, ReBase uses the instruction and few-shot examples provided by the new task to retrieve the relevant items from the datastore. Finally, ReBase uses an LLM to generate new data for the target task from the retrieved data.

be very similar, reducing performance of the fine-tuned models (Ye et al., 2022b;a). In response to these challenges, we propose **Re**trieval **Base**d Distillation (ReBase). As shown in Figure 1, ReBase is a framework that first retrieves data from an abundant and reliable labeled data source, then transforms them into the content and format necessary for the user's task. This data is then used to train a domain-specific model. Initially, ReBase scrapes online data and encodes them into a large datastore; Then, ReBase uses the user's instruction and the user's provided examples to retrieve the most relevant items from the large datastore. Finally, using an LLM, ReBase transforms the retrieved data point into a data that contains a query and an answer field for the specific task, this includes transforming the content and transforming the format. Different from previous methods, ReBase can effectively retrieve data from multiple dataset sources, enhancing the data's content diversity and avoids the issue where one or a few datasets do not contain sufficient information to fulfill the task's requirements. Moreover, ReBase adds a Chain-of-Thought transformation phase (Wei et al., 2022) where the LLM transforms the output into a step-by-step reasoning. This enables the small model to be trained on the reasoning generation by the large model, which is especially useful for reasoning tasks (Suzgun et al., 2022).

We test ReBase on a variety of benchmarks, including the BBH (Suzgun et al., 2022) benchmark, the MNLI (Williams et al., 2018) benchmark, SQuAD (Rajpurkar et al., 2016), and MCoNaLa code generation (Wang et al., 2023b). We found that ReBase improves the performance on BBH for **1.94%**, on SQuAD for **7.8%**, and on MNLI for **1.37%** over previous methods. Our method suggests the benefit of using data retrieved from multiple sources to train a specific model.

## 2 PROBLEM FORMULATION

We formulate the problem as follows: **Input:** The input contains an instruction of a task and few-shot examples. **Output:** The output contains a new dataset with the field (query, answer) that could be used to directly finetune a model. It also contains a task-expert model trained for this task. **Objective:** Our high-level objective is to generate a high-quality dataset that effectively boosts a model's performance on this task. Specifically, we assume that we have access to the abundant existing datasets online and access to LLMs. Our goal is to effectively harness the ability of LLMs and use the rich content of the existing datasets to create a high-quality dataset for the new task. Then use this dataset to train a task-expert model.

## 3 METHOD

In this section, we introduce the steps of ReBase: datastore construction, datastore retrieval, and dataset transformation. An overview of our method pipeline is shown in Figure 2.

## 3.1 DATASTORE CONSTRUCTION

Our datastore construction process begins with collecting datasets from Hugging Face Datasets (Lhoest et al., 2021), which consists of over 75,000 datasets. A Hugging Face dataset contains a dataset description that describes the purpose of the dataset. It also contains multiple rows entries and columns. Each row represents a data entry, and each column represents a specific attribute of that data entry. (eg. row_id, content, source_url, label)

For each row in these datasets, we do not directly encode the entire row entry because some attributes are redundant and may introduce noise (eg. attributes such as `row_id` or `url` are often not useful.) Instead, we encode each column separately. Specifically, for the $j$th row entry in dataset $i$, we iterate through each column $c$ in the row entry and encode it into a vector:

$$v_{i,j,c} = \texttt{Encode}\,(\texttt{column\_value})\,.$$

This vector has a unique identifier in the format:

$$\{\texttt{dataset\_id}, \texttt{row\_num}, \texttt{col\_name}\}$$

We then add the key-value pair $((i, j, c), v_{i,j,c})$ to the datastore. Additionally, for each dataset $i$, we encode its corresponding dataset description:

$$v_i = \texttt{Encode}\,(\texttt{dataset\_description})\,.$$

This value is identified by the dataset id $i$. We put the key-value pair $((i), v_i)$ into the datastore.

## 3.2 DATASTORE RETRIEVAL

In the datastore retrieval phase, our goal is to find relevant data across the different datasets. This process involves several steps to ensure the selection of the most relevant data.

First, we encode the user-provided instructions into $v_I$ using the same encoder used for the datastore. Then, we encode the user-provided examples. Each example should contain two fields: The query $q$ and the answer $ans$. We encode them separately into $v_q$ and $v_{ans}$.

Then, for each item $v_{i,j,c}$ in the datastore, we compute a cosine similarity between $v_q$ and $v_{i,j,c}$ to obtain a query score $\mathrm{S}_{\text{query}}^{(i,j,c)}$ for the item $(i, j, c)$. Similarly, we compute a cosine similarity between $v_{ans}$ and $v_{i,j,c}$ to obtain an answer score $\mathrm{S}_{\text{ans}}^{(i,j,c)}$ for the key $(i, j, c)$. If the user provides multiple examples, denote $Q_{\text{query}}$ and $Q_{\text{ans}}$ as the sets of encoded vectors for all user-provided query and answer examples, respectively. Then, for each item $v_{i,j,c}$ in the datastore, the query and answer scores for the key $(i, j, c)$ are calculated as:

$$\mathrm{S}_{\text{query}}^{(i,j,c)} = \frac{1}{|Q_{\text{query}}|} \sum_{q \in Q_{\text{query}}} \text{cos\_sim}(q, v_{i,j,c})$$

$$\mathrm{S}_{\text{ans}}^{(i,j,c)} = \frac{1}{|Q_{\text{ans}}|} \sum_{q \in Q_{\text{ans}}} \text{cos\_sim}(q, v_{i,j,c})$$

Next, for each row $(i, j)$, we define the query score and answer score for the row entry as the maximum query and answer scores across all columns:

$$\mathrm{S}_{\text{query}}^{(i,j)} = \max_c \mathrm{S}_{\text{query}}^{(i,j,c)}$$

$$\mathrm{S}_{\text{ans}}^{(i,j)} = \max_c \mathrm{S}_{\text{ans}}^{(i,j,c)}$$

Additionally, for each dataset $i$, we calculate a dataset score based on the cosine similarity between the encoded dataset description $v_i$ and the encoded task instruction $v_I$:

$$\mathrm{S}_{\text{dataset}}^{(i)} = \text{cos\_sim}(v_i, v_I)$$

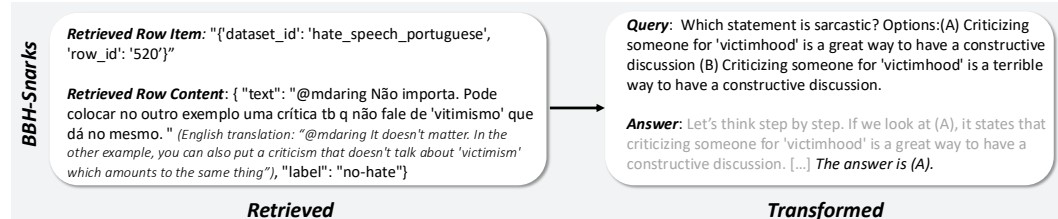

Figure 3: **Examples of ReBase transformations on BBH.** In the data transformation stage, ReBase takes in the original full row of the retrieved data and use the content to generate a new data with the field query and answer. The LLM need to identify the necessary fields in the row. For the BBH task, the transformation contains chain-of-thought reasoning.

The final score for each row $(i, j)$ in the datastore is calculated as the average of its query score, answer score, and dataset score:

$$S_{\text{final}}^{(i,j)} = \frac{1}{3}(S_{\text{query}}^{(i,j)} + S_{\text{ans}}^{(i,j)} + S_{\text{dataset}}^{(i)})$$

Finally, we sort all rows $(i, j)$ based on their final scores in descending order and select the top $N$ items with the highest scores. Using the selected $(i, j)$ identifiers, we query the original $j$th row in dataset $i$ and retrieve the original rows entry containing all the columns. This approach ensures that the selected data is highly relevant to the user's task, considering both the alignment on the user provided examples and the overall dataset context.

### 3.3 DATA TRANSFORMATION

After retrieving the relevant data, we employ a large language model (LLM) to transform the data into a format and content suitable for the specific task. This transformation process includes the following steps: **1. Salient Field Classification:** The LLM identifies the relevant fields in each retrieved row based on the domain-specific requirements. **2. Content Adaptation:** The LLM transforms the content to align with the target domain, ensuring it meets the specific needs of the task. **3. Chain-of-Thought (CoT) Generation:** For reasoning-intensive tasks, the LLM generates outputs using CoT, providing detailed step-by-step reasoning to enhance the quality and accuracy of the transformed data.

In our experiments, we use Claude 3 Haiku (Anthropic., 2024) as the LLM underlying the dataset transformer due to its competitive performance / cost tradeoff. The detailed prompt used to instruct the LLM is provided in the Appendix B. For tasks that require complex reasoning, such as the BIG-Bench Hard tasks, previous works have shown that Chain-of-Thought (CoT) (Wei et al., 2022) reasoning can greatly improve the model's performance on reasoning tasks (Suzgun et al., 2022) and finetuning on CoT data can further boost the reasoning ability (Chung et al., 2024) and can distill the reasoning capacity in LLMs to smaller models (Ho et al., 2022). Therefore, we leverage Chain-of-Thought generation. For these tasks, we prompt the LLM to generate a CoT reasoning followed by the final for the answer part instead of directly generating the final answer. The generated CoT data is then used for further training to improve the downstream model's performance as well. We demonstrate the transformation process in Figure 3.

Our transformation approach ensures that the transformed data is tailored to the new task in terms of both content and format and can be directly used for further finetuning. This process also incorporates the reasoning process of LLMs and distills such reasoning capacities to the task expert model.

## 4 EXPERIMENTS

In this section, we present our experiment settings, experiment results, analysis, and ablations.

Table 1: **Main quantatitive results.** We test on the MNLI, MCoNaLa, SQuAD, and BBH benchmarks. We also report the BBH-NLP and BBH-Algorithm which contains different subsets of BBH. We found that training on ReBase transformed data attains the best performance across theses tasks.

| Model | Data | MNLI | MCoNaLa | SQuAD(EM) | SQuAD(F1) | BBH | BBH-NLP | BBH-Alg |
|---|---|---|---|---|---|---|---|---|
| Retrieved | Prompt2Model | - | 13.1 | 50.5 | 63.0 | - | - | - |
| Claude-Haiku | 3-shot Prompting | 35.15 | 18.0 | 4.8 | 7.5 | 73.7 | - | - |
| GPT-4 | 3-shot Prompting | 87.81 | 41.6 | 74.3 | 87.1 | 83.1 | - | - |
| Llama3-8B | 3-shot Prompting | 44.4 | 28.4 | 43.2 | 54.1 | 56.8 | 65.3 | 50.0 |
| Llama3-8B | ZeroGen | 67.7 | - | 8.0 | 28.0 | - | - | - |
| Llama3-8B | Prompt2Model | 72.9 | 37.0 | 50.3 | 63.1 | 65.0 | 68.1 | 62.5 |
| Llama3-8B | ReBase | **74.3** | **38.2** | **58.1** | **71.7** | **66.9** | **69.5** | **64.9** |

## 4.1 EXPERIMENT SETTINGS

**Datasets** The datasets we used in this work include: **(i) MultiNLI (MNLI)** (Williams et al., 2018) tests the model's ability to recognize textual entailment between two sentences. It is one of the largest corpora for natural language inference, containing 433k samples across 10 distinct domains. We chose this task to test the method's performance on traditional language understanding. **(ii) SQuAD** (Rajpurkar et al., 2016) is a reading comprehension dataset that contains questions and context based on Wikipedia articles. We choose this task as another standard language understanding task. **(iii) MCoNaLa** (Wang et al., 2023b) is a multilingual benchmark to test models' ability to generate code from multi-lingual natural language intents. We focus on the Japanese-to-Python subtask, as it is a challenging task with no task-specific annotated data available. **(IV) BIG-Bench Hard (BBH)** (Suzgun et al., 2022) is a challenging reasoning benchmark. It is a subset of BIG-Bench (BIG-bench Authors, 2023) containing challenging tasks where LLMs underperform humans. This dataset tests whether ReBase can generate data for highly challenging reasoning tasks.

**Baselines** **(1) Prompt2Model** (Viswanathan et al., 2023a) This method retrieves a model from Hugging Face via the task instruction, then finetunes this model using both synthesized and retrieved datasets (without transforming the latter). **(2) Synthesized Data** We use the dataset generation method described by Prompt2Model (Viswanathan et al., 2023a) to obtain synthesized data and use it to finetune a LLM. This generation process uses dynamic temperature and prompt sampling to increase the synthesized data's diversity and demonstrates impressive data synthesize ability. **(3) ZeroGen** This method uses pretrained LLMs to directly generate datasets under zero-shot setting. This method is targeted for classification tasks. We use Claude-Haiku to generate the data. **(4) Few-Shot Prompting** For this, we directly prompt the pretrained LLM with few-shot examples without any finetuning. We report Claude Haiku which is used as our dataset generator and transformer. We also report GPT-4 as a strong upper bound model.

**Implementation Details** We use a pretrained model[2] from the Sentence Transformers toolkit (Reimers & Gurevych, 2019) to encode all data in the datastore construction phase. We use 3K examples for MNLI and SQuAD and 1K for MCoNaLa and each BBH task. We use Claude 3 Haiku model to transform the data. To more accurately simulate the case in which we are tackling a new task without training data, we prevent the retriever from retrieving any data from the target task's original training set. For model training, we choose the most recent open-source LLM Llama3-8B (AI@Meta, 2024) as the base model for both the synthesized method and ReBase. We train the model using QLoRA (Dettmers et al., 2023) which requires only one NVIDIA A6000 48GB GPU. We provide training details in Appendix D

**Metrics** We report ChrF++ (Popović, 2015) score for MCoNaLa, this metric was similarly used for evaluation by (Viswanathan et al., 2023a). For MNLI, we report the accuracy. For SQuAD, we use same exact match metric and F1 metric in (Rajpurkar et al., 2016). For BBH, we use the evaluation script from (Yue et al., 2023) to first extract the answer in the generated sentence and then report the accuracy.

---

[2] `distiluse-base-multilingual-cased`

## 4.2 RESULTS

**Quantitative Results** We present our main results in Table 1. For MNLI, BBH, SQuAD, and MCoNaLa ReBase outperforms the data synthesis method by 1.37%, 1.94%, 7.8%, 1.2% respectively. Specifically on BBH, ReBase outperforms by 1.39% on the BBH-NLP split and 2.37% on the BBH-Alg split. On the question answering benchmark SQuAD, ReBase outperforms synthesized method by 7.8%. These results demonstrate the ReBase's effectiveness by retrieving then transforming the data compared with directly generating all the data using LLM.

**Qualitative Results** We present the qualitative results in Figure 4 to demonstrate the data obtained through ReBase and the data obtained through synthesized method in the MCoNaLa benchmark and SQuAD benchmark. In MCoNaLa, the task is to generate data with a Japanese instruction as input and a corresponding python program as output. We found that ReBase outputs data samples that contains more programs with higher diversity and programs that require more complicated reasoning process such as dynamic programming whereas synthesized method only gives simple instructions that require a few lines of codes. In SQuAD, the task is to generate data with a question and a context as input and an answer to the question as output. We found that ReBase greatly increases the question diversity in terms of content and creates questions that require more complicated reasoning where as the synthesized data only asks questions that are simpler, more well known, and more straightforward. Interestingly, we found that ReBase does not increase the length of the context part in the data compared with synthesized data. We provide more results in Appendix F.

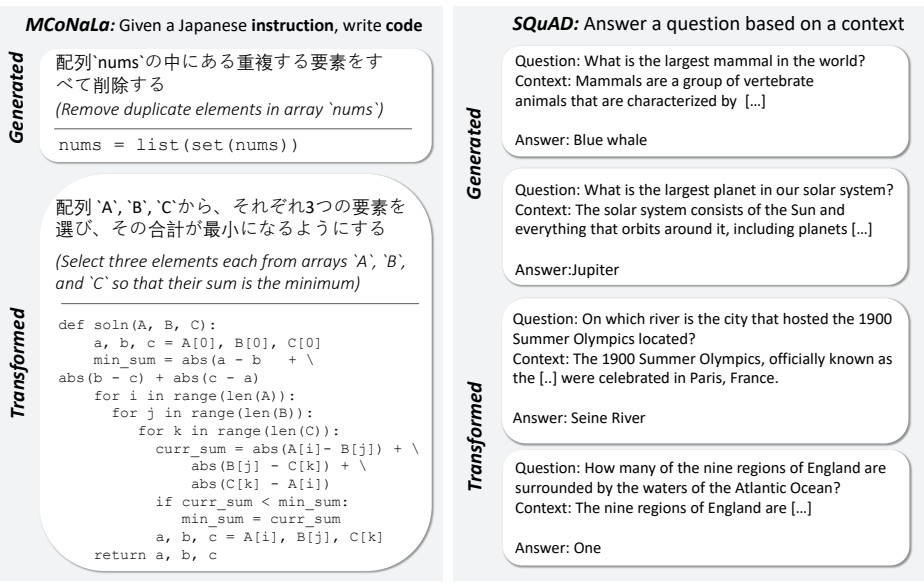

Figure 4: **Qualitative Examples on ReBase (Transformed) compared to directly synthesized data (Generated).** ReBase outputs more diverse data while directly synthesized data tend to be simpler and replicate. In MCoNaLa, ReBase generates samples that contains dynamic programming, counting, mathematical calculations whereas directly synthesized dataset is limited to simpler commands such as printing or simple list operation. In SQuAD, we found that ReBase generates samples that contain diverse and harder logics whereas directly synthesized data asks simple facts.

## 4.3 ANALYSIS

**Dataset Source** One of the benefits of constructing the database is that the model can retrieve from multiple dataset sources to get the relevant items from each of them. To analysis how this effects the data for each task, we analyzed the number of different datasets in its retrieved data for each task. We present the result in Table 2. The results demonstrate that all the tasks retrieves from at least 20 different dataset sources. MCoNaLa and SQuAD retrieves from more than 50 different datasets.

BBH tasks retrieves from 35 datasets on average. MNLI retrieves from 20 datasets. We provide a more detailed data source analysis in Appendix A.

Table 2: **Dataset source analysis.** For datasets generated by ReBase, we calculate the number of unique datasets that it retrieves from. Results show that each benchmark above retrieves from at least 20 different datasets. Detailed information is in Appendix A.

| Benchmark | MCoNaLa | MNLI | SQuAD | BBH (total) | *BBH-NLP* | *BBH-Alg* |
|---|---|---|---|---|---|---|
| **# of Sources** | 67 | 20 | 55 | 35 | 36 | 46 |

**Dataset Diversity**  Previous works have shown that synthesized data lacks in diversity (Ye et al., 2022a) and sometimes produces near-duplicate samples (Gandhi et al., 2024). We study whether ReBase increases the datasets' diversity. We follow DataTune (Gandhi et al., 2024) to conduct diversity analysis on MCoNaLa, MNLI, and SQuAD. First, we calculate the uniqueness of the dataset samples on these three benchmarks. We use ROUGE-L (Lin, 2004) to determine whether a sentence is unique in the dataset (Wang et al., 2022). Specifically, for a sentence $s$, if the ROUGE-L score between $s$ and every other sentence $s'$ is smaller than a threshold $T$, we decide this sentence to be unique. In our experiment, we use the threshhold 0.7. The results are shown in the Unique Percentage column of Table 3, we found that ReBase significantly increases the percentage of unique samples in the dataset compared with synthesized data. The synthesized data yields less than 50% of non-duplicate samples across the three benchmarks, while ReBase results in more than 70% non-duplicate samples across the three benchmarks. We also calculate the average unique unigrams, and unique bigrams per created example to measure the lexical difference. The results are demonstrated in Table 3. ReBase significantly increases the average unique unigrams and bigrams on the three benchmarks.

**Embedding Visualization**  We conduct embedding visualization on SQuAD and MNLI to visualize the datasets. We use MiniLM v2 (Wang et al., 2021) to encode each sentence and then project the embeddings into a 2D space using t-SNE (van der Maaten & Hinton, 2008). The results are shown in Figure 5. We found that the data generated by ReBase are more widely scattered across the embedding space compared to the synthesized data, which have smaller coverage. Additionally, we observed that the total coverage of ReBase and synthesized data is greater, indicating the potential for further combining ReBase and synthesized data to create a more powerful dataset.

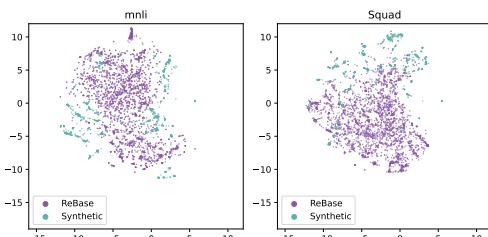

Figure 5: **Embedding Visualization Result of MNLI and SQuAD.** The data generated by ReBase are more widely scattered across the embedding space compared to the synthesized data.

### 4.4 ABLATIONS

**Ablations on Filtering**  We noticed that for some tasks that are not associated with very relevant documents in the datastore, the transformed data contains noise that may impair the data quality. Training on such data may reduce the performance and make the model underperform the pretrained model. Therefore, we conduct experiments on using an LLM as a filterer and filter out the data that doesn't comply to the format or contains irrelevant noise in the content. The detailed prompt used to instruct the LLM is provided in the Appendix B. We use GPT-3.5-turbo as the filterer and then use the filtered data to train Llama3-8B on the 27 tasks on BBH and MCoNaLA, the results are shown in Table 4. We found that filtering doesn't increase the overall performance on BBH

Table 3: **Dataset Diversity Analysis.** We report the data uniqueness percentage, the average unique unigrams and unique bigrams per sample. We found that ReBase significantly increases the number of average unique unigrams, average unique bigrams, and the unique percentage of the dataset, suggesting the ReBase promotes data diversity in the dataset.

| Task | Method | Unique Unigrams | Unique Bigrams | Unique Percent |
|------|--------|-----------------|----------------|----------------|
| MCo NaLa | Syn | 0.56 | 0.36 | 25.90% |
| | ReBase | 1.85 | 1.99 | 75.42% |
| MNLI | Syn | 0.62 | 2.00 | 21.61% |
| | ReBase | 3.28 | 12.21 | 71.05% |
| SQuAD | Syn | 2.20 | 10.94 | 37.69% |
| | ReBase | 6.31 | 29.33 | 96.56% |

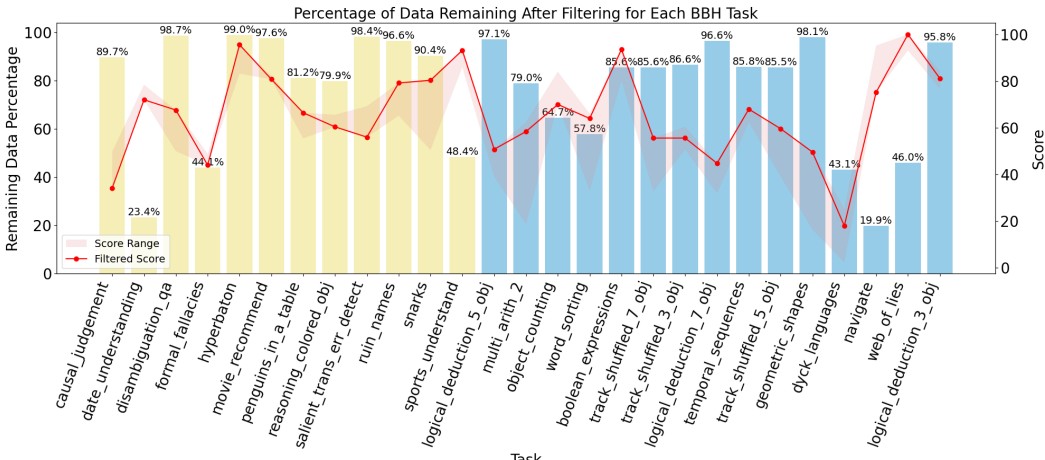

Figure 6: The bars represent the percentage of remaining data after filtering for each BBH task. The shaded area in the figure indicates the range of pretrained scores, transformed scores, and filtered data training scores for each task. The full names of the abbreviated task names are in Appendix E

and MCoNaLa. While filtering can enhance performance on certain tasks where training on ReBase harms performance, it decreases performance on others. Such performance drop is potentially due to the decrease in dataset size. Figure 6 shows the percentage of remaining data after filtering for each BBH task and the effect of filtering on the scores. We provide details on filtering in Appendix C.

**Ablating on Data Size** In our experiments, we use a data size of 1k for both ReBase and synthesized data. In this experiment, we study the effect of data size by varying the amount of data we use to train the model. Specifically, we vary the data size by 200, 400, 600, 800, and 1000 and then test on BBH. For experiment on dataset size K, we use the retrieved data with the top K highest scores. We report the results in Table 5. The results show that using 1k data achieves the best performance. In general, scaling up the dataset size enhances the performance. This highlights the importance of obtaining adequate data for a given task.

**Ablating the Data Generation Model** In out experiments, up to this point we have mainly used Claude 3 Haiku (Anthropic., 2024) for the transformation and data synthesis. In this experiment, we test the effect of using a different, more expensive model, GPT-4, instead. We use data size 1k for MCoNala and 200 for BBH and report the performance in Table 6. For MCoNaLa, interestingly, GPT-4 significantly outperforms Haiku with synthesized data, but with ReBase the gap closes significantly, demonstrating that ReBase may allow more computationally efficient models to serve as teachers for data distillation. In fact, Haiku with ReBase outperforms GPT-4 without ReBase, at nearly two orders of magnitude less cost. For BBH, we found that GPT-4 with synthesized data outperforms ReBase whereas when using Claude 3 Haiku, synthesized data underperforms ReBase.

Table 4: **Results of the filtered ReBase dataset.** We use the filtered dataset to test on BBH, BBH-NLP, BBH-Alg, and MCoNaLa. We found that filtering does not increase the overall performance on three benchmarks, suggesting that dataset size, in addition to noise, also impacts performance. We provide detailed illustration of the BBH tasks in Figure 6 and further discussion in Appendix C.

|  | BBH | BBH-NLP | BBH-Alg | MCoNaLa |
|---|---|---|---|---|
| Filtered | 65.71 | 69.15 | 62.96 | 37.24 |
| ReBase | **66.90** | **69.45** | **64.85** | **38.24** |

Table 5: **Results on using different dataset size on the BBH benchmark.** Generally, increasing the dataset size boosts performance, suggesting the importance of obtaining adequate data for a task.

| Data Size | BBH | BBH-NLP | BBH-Alg |
|---|---|---|---|
| 200 | 59.19 | 61.17 | 57.60 |
| 400 | 64.70 | 68.36 | 61.76 |
| 600 | 62.40 | 65.36 | 60.03 |
| 800 | 65.65 | 68.52 | 63.36 |
| 1000 | **66.90** | **69.45** | **64.85** |

This shows that ReBase may be useful to better unleash the CoT reasoning ability of cheaper models, but less effective in further promoting the CoT reasoning of expensive and powerful models.

Table 6: **Ablation on the LLM used on the MCoNaLa and BBH task.** We conduct experiments on using GPT-4 and Claude 3 Haiku on MCoNaLa and report ChrF++ score, and on BBH using accuracy. Using the more powerful GPT-4 model boosts performance for both the synthesized dataset and ReBase on both datasets, but also costs 100 times more than using the Claude 3 Haiku model.

|  | Method | GPT-4 | | Claude3-Haiku | |
|---|---|---|---|---|---|
|  |  | Acc | Cost | Acc | Cost |
| MCo--NaLa | Syn | 37.88 | $9.53 | 36.98 | $0.11 |
|  | ReBase | **38.48** | $8.03 | **38.24** | $0.11 |
| BBH | Syn | **65.43** | - | 57.22 | - |
|  | ReBase | 64.95 | - | **59.19** | - |

**Ablating on Retrieval Score**    We provide ablation analysis on the retrieval method. In ReBase, we use the average score of the input, output, and dataset similarity. In this ablation, we tried to (1) use the dataset score only and (2) use the average of the output score and the input score. The result is shown in Table 7, we found that using the average of the three scores attains the best performance.

**Ablating on Chain-of-Thought**    We conduct ablation experiment on CoT by running Prompt2Model w/ CoT synthesized data (etc. using both directly retrieved data and also synthesized CoT data) and running ReBase without CoT. The result is shown in Table 8. We found that Prompt2Model w/ CoT under-performs ReBase, this is likely due to that for BBH tasks, the retrieved data have a large domain gap with the target task, and using directly retrieved data would introduce noise in the training phase, thus reducing the performance. ReBase w/o CoT underperforms the other methods w/ CoT. This aligns with previous findings that CoT distillation helps performance on BBH reasoning tasks. It also suggests that ReBase is compatible with the CoT distillation method.

**Ablating on Domain Gap**    We conduct experiment on introducing different levels of domain shifts. On MCoNaLa, we manually delete the top-1 relevant dataset (the dataset with the highest dataset score) during retrieval and deleting the top-2 relevant dataset. The result is shown in Table 9. We found that the performance drops as the domain gap increases. This suggests that it is easier for the model to transform data of similar domain into the target domain.

Table 7: **Ablation Result on the retrieval method.** We use different retrieval methods w/ Re-Base on the MCoNaLa benchmark. We find that retrieving with all the three scores attains the best performance.

|  | Dataset Score | Input-Output Score | ReBase |
|---|---|---|---|
| MCoNaLa | 19.20 | 24.30 | **38.24** |

Table 8: **Ablation Results on CoT.** We conduct experiments of ReBase w/o CoT, ReBase w/ CoT and Prompt2Model w/ CoT on three BBH benchmarks. We found that ReBase w/ CoT performs the best, indicating that ReBase is compatible with CoT distillation and that CoT helps performance on retrieval based distillation.

| Method | Boolean Expression | Date Understanding | Object Counting |
|---|---|---|---|
| ReBase w/o CoT | 68.0 | - | |
| Prompt2model w/ CoT | 83.2 | 53.2 | 57.6 |
| ReBase w/ CoT | **94.0** | **77.6** | **72.0** |

Table 9: **Ablation Result on Domain Gap.** We conduct experiment by deleting the most relevant dataset/deleting the top-2 most relevant dataset during retrieval. On MCoNaLa, The performance drops as the domain gap increases, suggesting that it is easier to for the model to transform data from similar domains into the target domain.

|  | All Domain | Del Top 1 | Del Top 2 |
|---|---|---|---|
| MCoNaLa | 38.24 | 36.71 | 35.14 |

## 5 RELATED WORK

**Retrieval-Augmented Generation (RAG)** Retrieval-Augmented Generation (RAG) (Lewis et al., 2020; Gao et al., 2024; Asai et al., 2023; Chen et al., 2017) retrieves from external knowledge to help the LLM answer open-domain questions. Recent works demonstrate that RAG can greatly boost the reasoning ability of LLMs (Jiang et al., 2023; Shao et al., 2023). IAG (Zhang et al., 2023) leverages both retrieved knowledge and inductive knowledge derived from LLMs to answer open-domain questions. Inspired by the success of RAG, we study how retrieving from external knowledge improves dataset quality and further improves model performance.

**Data Synthesis** Recent studies use LLMs as dataset generators (Patel et al., 2024; Song et al., 2024) and focus on improving the generated data's quality. Zerogen (Ye et al., 2022b) uses pretrained LLMs to generate datasets directly under zero-shot setting. Progen (Ye et al., 2022a), Sungen (Gao et al., 2023), and Impossible Distillation (Jung et al., 2024) uses feedback from smaller models to distill the generated data. AttrPrompt (Yu et al., 2023a) improves data quality by improving the prompt. Unnatural Instructions (Honovich et al., 2022), ReGen (Yu et al., 2023b), and S3 (Wang et al., 2023a) improves the data quality by using other datasets as reference. We explores the use of both RAG and LLM's generation ability to create a diverse and reliable dataset for specific tasks.

## 6 CONCLUSION

In this paper, we present ReBase, a framework that uses retrieval and transformation to create diverse and high-quality domain-specific dataset to train task-expert models. Our method shows significant improvement over conventional dataset generation methods. We establish the benefit of leveraging examples retrieved from a large, heterogenous datastore to create task-specific training data. We believe this work motivates future work on retrieving labeled examples from a prompt; improved example retrieval could lead to significantly improved retrieval-based distillation.

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

## A    BBH DATA SOURCE DETAILS

In this section, we provide a detailed analysis of BBH tasks dataset source. In the main text, we report the number of different data sources (the number of distince (dataset, dataset_config) pairs) that each task retrieves from. In this part, we report the number of different datasets. We report the average of all the BBH tasks and present the statistics in Table 11. In Figure 7, we demonstrate the number of data sources for each BBH task. We found that most tasks retrieves from 30 data sources. Object Counting and Word Counting retrieves from up to 120 data sources while Boolean Expressions retrieves from 4 data sources. This suggests that the number of dataset sources can greatly vary depending on the task type.

Table 10: **BBH task abbreviation clarification.** We show the mapping between the original BBH task name and the abbreviation that we used in our paper.

| Task Name | Abbreviation |
|---|---|
| **multistep_arithmetic_two** | multi_arith_2 |
| **salient_translation_error_detection** | salient_trans_err_detect |
| **tracking_shuffled_objects_three_objects** | track_shuffled_3_obj |
| **tracking_shuffled_objects_five_objects** | track_shuffled_5_obj |
| **tracking_shuffled_objects_seven_objects** | track_shuffled_7_obj |
| **logical_deduction_three_objects** | logical_deduction_3_obj |
| **logical_deduction_five_objects** | logical_deduction_5_obj |
| **logical_deduction_seven_objects** | logical_deduction_7_obj |

Table 11: **Detailed BBH dataset source.** We also report the number of unique datasets for each task. On a dataset level, the BBH retrieves from 24 different datasets on average, suggesting that the retrieved data comes from very diverse sources.

| Task | # of Dataset | # of Dataset Source |
|---|---|---|
| BBH (total) | 24 | 42 |
| *BBH-NLP* | *21* | *36* |
| *BBH-Alg* | *27* | *46* |

## B    PROMPTS

We present the prompt that we used to transform a retrieved row entry and the prompt we used to filter the data.

### B.1    TRANSFORM PROMPT

"' I would like you to create questions for a test. The directions for the test are:

'''

{task_description}

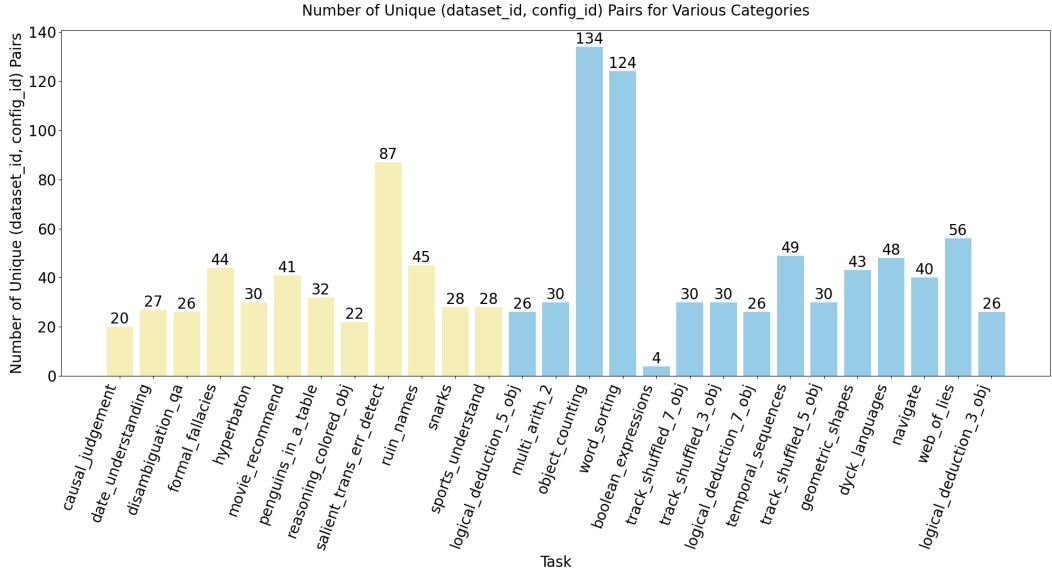

Figure 7: **The number of Dataset Sources for each BBH task.** The bars represent the number of unique data sources retrieved for BBH tasks (This is calculated as the number of unique (dataset, config) pairs of the retrieved data). We found that most BBH tasks retrieve data from around 30 sources, demonstrating the diversity data source of ReBase. Among the BBH tasks, Object Counting and Word Sorting retrieves from more than 120 sources while Boolean Expression retrieves from only 4 sources. The suggests that the amount of dataset sources is largely relevant to the task.

''' 

The format should be in json like this:

{example}

Now I will provide you with a JSON file from a different dataset. Please create a question where the format and type of question is similar to the examples provided above, but the content is inspired by the example provided below. You need to decide which part of the dataset to use.

{dataset_row}

Your response MUST be a JSON with exactly 2 fields: "input" and "output".
Response (JSON ONLY): '''

## B.2 FILTER PROMPT

''' You will be given a task description. Your task is to determine whether a data is fitful for this task.
# Instruction:

{task_description}

# Fitful Examples that meet the task's request:

{example}

Now, there is a new data. Your task is to determine whether this data is fitful for this task.
New Data:

```
{{
"input": "{input_data}",
"output": "{output_data}",
}}
```

Response (Yes or No): '''

## C    ABLATION ON FILTERING

### C.1    PIPELINE

A filter pipeline is demonstrated in Figure 8 where the LLM filters out the samples that contain noise or are unanswerable given the task instruction and few-shot examples.

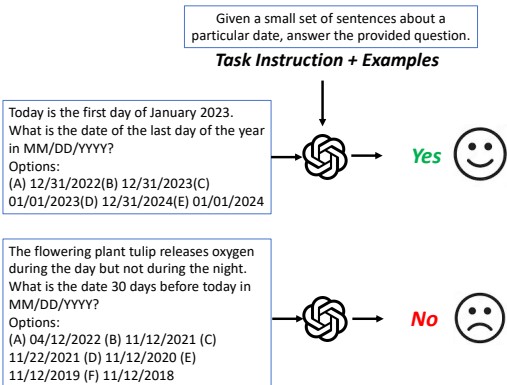

Figure 8: **Filter Pipeline.** We instruct the LLM to filter with task instruction and few examples. Then, we input the current example to the model and let the model choose whether the current example can be used to train a model for the task.

### C.2    ANALYSIS

We observed that most tasks maintain a high percentage of data after filtering. Most tasks retain over 80% or even 90% of the original data. This suggests that ReBase transformed data is generally plausible and usable for downstream finetuning and the filtering process does not substantially reduce the dataset size. However, there are some exceptions. For *date_understanding*, *formal_fallacies*, *sports_understanding*, *dyck_languages*, *navigate*, and *web_of_lies*, the percentage of the remaining data drops below 50% or even under 20%.

We observed that filtering can be beneficial in certain cases but not always. When the filtering removes a large amount of data, performance tends to decline. For instance, tasks such as *date_understanding*, *formal_fallacies*, *dyck_languages*, and *navigate* decline after filtering. However, *sports_understanding* shows improvement in performance after filtering nearly 50% of the data.

## D    TRAINING DETAILS

In our experiments, we use QLoRA to train meta-llama/Meta-Llama-3-8B for 1 epoch using a learning rate of 3e-4, a batch size of 2 per device, warmup steps of 20, and gradient accumulation steps of 4. We use 8-bit AdamW optimizer with a weight decay of 0.001 and a linear learning rate scheduler.

## E    BBH TASK ABBREVIATION

Due to the length of some task names, abbreviations are used in the figure. The full names can be found in Table 10.

## F    ADDITIONAL QUALITATIVE RESULTS

In Figure 9, we show more examples of the data generated by ReBase and the synthesized data. We found that ReBase generates data that contains complicated math calculaitons and dynmaic programming. Whereas synthesized data is limited to simple operations.

**MCoNaLa:** Given a Japanese **instruction**, write **code**

*Generated*

文字列 'hello' を3回繰り返して表示する

*(Display the string 'hello' repeatedly three times)*

```
print('hello' * 3)
```

*Transformed*

整数 n, a, b, c を受け取り、n を a, b, c の長さの最大の分割数に分割する関数を作成する

*(Write a function that takes integers n, a, b, c and divides n into the largest number of divisions of length a, b, c.)*

```
def maximum_segments(n, a, b, c):
    dp = [0] * (n + 1)
    for i in range(1, n + 1):
        dp[i] = max(dp[i - a] + 1 if i >= a else 0,
                    dp[i - b] + 1 if i >= b else 0,
                    dp[i - c] + 1 if i >= c else 0)
    return dp[n]
```

**MCoNaLa:** Given a Japanese **instruction**, write **code**

*Generated*

ファイル'sample.txt'の内容を行ごとに読み込み、各行の単語数を表示する

*(Read the file 'sample.txt' line by line and display the number of words in each line.)*

```
with open('sample.txt', 'r') as f:
    for line in f:
        print(len(line.split()))
```

*Transformed*

nCr mod pの値を計算する関数を書いてください

*(Write a function to calculate the value of nCr mod p)*

```
def nCr_mod_p(n, r, p):
    if (r > n - r):
        r = n - r
    C = [0 for i in range(r + 1)]
    C[0] = 1
    for i in range(1, n + 1):
        for j in range(min(i, r), 0, -1):
            C[j] = (C[j] + C[j-1]) % p
    return C[r]
```

Figure 9: **Additional Qualitative Examples on ReBase compared to directly synthesized data.** In MCoNaLa, ReBase outputs math modula and dynamic programming programs whereas synthesized method is limited to simple operations.

