# OpenReview forum: "Training Task Experts through Retrieval Based Distillation"
_ICLR.cc/2025/Conference — ICLR 2025 Conference Withdrawn Submission_

### Official Review · Reviewer_XCzu · 2024-11-01

**Soundness:** 2
**Presentation:** 3
**Contribution:** 2
**Rating:** 5
**Confidence:** 4

**Summary:**

The paper introduces ReBase, a method retrieves from online sources and transforms them into sample-style datas. Using the online datas ,the method can enhances data diversity and also generates the CoT datas by distilling LLMs. The Experimental results show the method improves performance on SQuAD, MNLI and BigBench-Hard.

**Strengths:**

The motivation is practical and significant, offering a cost-effective solution for obtaining high quality domain-specific datas using online datas.

Paper demonstrates the performance improvement of the ReBase method in multiple benchmarks, such as achieving 7.8%, 1.37%, and 1.94% performance improvements on SQuAD, MNLI, and BigBench Hard datasets, respectively. These results demonstrate the effectiveness of the method.

**Weaknesses:**

Although ReBase performs well on specific benchmark tests, its generalization ability on other types of tasks has not been fully validated. Because the three datasets involved in the article are too simple and early, there is a risk of label exposure. Meanwhile, these three datasets are all generic datasets and do not match the domain specific mentioned in the article.

For  specific domain and unseen types of data, online data may not have similar data, and enhancing through retrieval may not bring real benefits.

Although the ReBase method improves data diversity, there is insufficient discussion in the article about the potential data bias and quality issues that may be introduced during retrieval and conversion processes

**Questions:**

How does the ReBase method generalize across different types of tasks and domains? Is there a plan to validate the effectiveness of this method on other types of tasks?

How to ensure data quality and reduce potential biases during the process of retrieving and converting data? Is there a mechanism to monitor and correct these deviations?

---

### Official Review · Reviewer_Pxv3 · 2024-11-02

**Soundness:** 3
**Presentation:** 3
**Contribution:** 3
**Rating:** 5
**Confidence:** 4

**Summary:**

The paper proposes a retrieval-based task-specific data construction method aimed at improving the quality of task-specific data generated by large language models (LLMs). Specifically, a set of databases was first constructed from Hugging Face. Then, a retrieval method was designed to obtain data based on instructions and examples. The retrieved data is subsequently transformed using LLMs and prompts to make it directly usable for training. The authors test their method on 4 benchmarks and results show that the method significantly improves performance by up to 7.8% on SQuAD, 1.37% on MNLI, and 1.94% on BigBench-Hard.

**Strengths:**

- Data Diversity: ReBase enhances data diversity by retrieving data from multiple online sources
- Reasoning Distillation: ReBase introduces a Chain-of-Thought transformation phase, allowing smaller models to be trained using the reasoning processes generated by larger models, which is particularly useful for reasoning tasks.
- Performance Improvement: In multiple benchmark tests, ReBase has demonstrated significant performance improvements compared to traditional data generation methods.

**Weaknesses:**

- The LLM used in the text is Llama3-8B. Have you tried any other models (different size of params, different model families) ?
- Processing large amounts of data into datastore and retrieve from them can be computational costly, esp. computing cosine similarities. Any efforts to make such computation less so?
- Have there been any assessments of the data quality after TRANSFORMATION, such as issues like hallucinations, the CoT is complete and correct, etc.?

**Questions:**

- Could the experiments be conducted on more baseline LLMs? so to verify its effectiveness on a broader scale (different model sizes, different model families)
- Propose some methods to improve the efficiency of the proposed methods.
- Evaluate the quality of the data after the transformation.

---

### Official Review · Reviewer_5o5m · 2024-11-04

**Soundness:** 2
**Presentation:** 2
**Contribution:** 2
**Rating:** 3
**Confidence:** 4

**Summary:**

A retrieval-augmented synthetic data generation framework for strong-to-weak distillation.

**Strengths:**

S1. The paper presents the efficacy of retrieval augmentation for new data generation.

S2. The proposed method demonstrates improvement over baseline methods.

**Weaknesses:**

W1. Missing important related work "Retrieval-Augmented Data Augmentation for Low-Resource Domain Tasks"[1].

W2. Usage of the HuggingFace datasets may cause contamination if not explicitly taken care of.

W3. The choice of teacher model is unclear as the ability varies for tasks such as coding or math. How is coding or math correctness assessed for the synthetic samples?

W4. It is vague how the data sources are selected.


[1] Seo, Minju, et al. "Retrieval-augmented data augmentation for low-resource domain tasks." arXiv preprint arXiv:2402.13482 (2024).

**Questions:**

Please see the weaknesses.

---

### Note · Authors · 2024-11-16

I have read and agree with the venue's withdrawal policy on behalf of myself and my co-authors.